# Non-communicable diseases and inequalities increase risk of death among COVID-19 patients in Mexico

Juan Pablo Gutierrez[1]*, Stefano M. Bertozzi[2,3,4]

1 Center for Policy, Population & Health Research, School of Medicine, National Autonomous University of Mexico, Mexico City, Mexico, 2 University of California, Berkeley, California, United States of America, 3 University of Washington, Seattle, Washington, United States of America, 4 National Institute of Public Health, Mexico (INSP), Cuernavaca, Mexico

* jpgutierrez@unam.mx

**Data Availability Statement:** Data is available at https://www.gob.mx/salud/documentos/datos-abiertos-152127.

**Funding:** The author(s) received no specific funding for this work.

## Abstract

### Background

The SARS-CoV-2 pandemic compounds Mexico's pre-existing challenges: very high levels of both non-communicable diseases (NCD) and social inequity.

### Methods and findings

Using data from national reporting of SARS-CoV-2 tested individuals, we estimated odds of hospitalization, intubation, and death based on pre-existing non-communicable diseases and socioeconomic indicators. We found that obesity, diabetes, and hypertension are positively associated with the three outcomes in a synergistic manner. The municipal poverty level is also positively associated with hospitalization and death.

### Conclusions

Mexico's response to COVID-19 is complicated by a synergistic double challenge: raging NCDs and extreme social inequity. The response to the current pandemic must take both into account both to be effective and to ensure that the burden of COVID-19 not falls disproportionately on those who are already disadvantaged.

## Introduction

The current SARS-CoV-2 pandemic has already infected more than 14 million individuals and caused over 600 thousand deaths worldwide [1]. There is an urgent need to better understand who is at higher risk of worse outcomes, including death. A recent study suggests that 22% of the global population is at risk of developing severe COVID-19 based on underlying conditions, principally non-communicable diseases (NCD); 4% of global population (included in the 22%) is at high-risk based on the same conditions [2]. Poverty is associated with increased

**Competing interests:** The authors have declared that no competing interests exist.

prevalence of NCDs (and with obesity, another risk factor not considered in the Clark *et al* study) but it also is associated with increased risk of infection, poorer control of NCDs and poorer access to quality health services, thus compounding the impact on the poor [3].

Previous studies have reported that obesity and diabetes are associated with greater COVID-19 disease severity [4–6], and that has also been documented in Mexico using data from the initial phase of the pandemic [7]. Other conditions that have been positively associated with the severity of COVID-19 –and that are also major contributors to the global burden of disease—[8] include: smoking, chronic cardiac disease, chronic pulmonary disease and chronic kidney disease [9, 10]. An analysis of COVID-19 cases in the United Kingdom found that male sex, older age, economic deprivation, uncontrolled diabetes, and asthma increased risk of death [11].

The reported prevalence of obesity in Mexico among individuals 15 years of age and older is 32.4%, the second highest in the OECD, behind only the USA. [12]. Data from the Mexican National Health and Nutrition Surveys describe the growth in prevalence of obesity since at least 2000; the growth is faster among younger individuals [13].

Diabetes is the single largest contributor to disease burden in Mexico, with a prevalence of about 15% of the adult population. Hypertension is also very important, with a prevalence of 30% [14–16].

Previous analyses have highlighted major inequalities in health in Mexico, as individuals with lower incomes face a higher probability of complications once diagnosed with a non-communicable disease (NCD) and an increased financial burden that affects their ability to adhere to treatment recommendations [17, 18]. Given COVID-19's association with NCDs, the fact that NCDs are both more prevalent and less well controlled among the poor, and the fact that the poor are at greater risk of infection because they are unable to shelter in place without income—the pandemic is expected to further worsen the health of the poor compared to those who are better off, increasing health inequity [3, 19–21].

In Mexico, the triad of NCDs, inequity and COVID doesn't occur in the context of a high-functioning health care system, but rather one that is characterized by fragmentation, under-funding, lax regulatory oversight and a high proportion of poorly-trained health personnel. The unfortunate result is low average health care quality which translates into unacceptably high mortality rates for severe COVID-19 disease.

Using data from SARS-CoV-2-positive individuals in Mexico, our aim is to assess NCDs and socioeconomic status as risk factors for COVID-19 severe outcomes, in particular hospitalization, intubation and death.

## Methods

We analyzed public data reporting all individuals tested for SARS-CoV-2 in Mexico compiled and published by the Directorate General of Epidemiology (Ministry of Health). It is updated daily and included basic demographics and comorbidities, in addition to the test result [22]. The reports also include place of residence and whether the individual died. We only included adults age 20 and older in our analyses as our interest is on whether NCDs and inequalities are related to worse outcomes among patients with COVID-19. The prevalence of NCDs and the incidence of severe COVID-19 disease are both very low among those under 20.

Testing for SARS-CoV-2 in Mexico at public facilities is only performed if the individual has clinical symptoms suggestive of COVID-19. Thus, not all individuals who ask for a test are tested, but individuals who were tested were included in the dataset, albeit with variable reporting delays. It is important to note that included cases represent both ambulatory and hospitalized individuals.

The dataset is comprised of cases from the national surveillance system in which 475 clinical facilities in the country test at least 10% of all cases with mild respiratory symptoms and 100% of cases with severe symptoms. Also included are cases with severe symptoms reported from all clinical facilities not included in the national surveillance system [23].

The dataset includes 35 variables: five variables related to identification of the case (date of the report, unique case identifier, whether the testing facility was a surveillance site or not, provider, state where the facility is located); ten variables capture the demographic characteristics of the individual (sex, age, state of birth, state and municipality [county] of residence, nationality, migration status, and, for imported cases, country of origin and native language); for woman there is a variable on current pregnancy; ten variables report pre-existing conditions obtained by individual self-report (diabetes, COPD, asthma, immunosuppression, hypertension, cardiovascular disease, obesity, chronic renal disease, smoking, and "other conditions"); two variable capture SARS-CoV-2 status (result of the test and if the test subject reported contact with another case); and six variables related to COVID-19 case management (reported date of first symptoms, if care was ambulatory or hospitalized, date of hospitalization—for those hospitalized, date of death, intubation, diagnosis of pneumonia, and if the patient required critical care [ICU]).

Underlying conditions reported in the dataset are reported by facilities in the country using their standard reporting procedures and did not include further detail on those conditions (such as severity or whether the diagnosis was confirmed).

For the analyses we used the variables as reported. We excluded in the regression analysis observations with missing values as those were a negligible fraction of the database, at less than 0.5% of observations. Overall, for the regression analysis, 3.5% of observations were excluded (23,037) due to missing values. The incidence of complications (hospitalization, intubation, death) among the excluded observations were similar to the included observations.

Descriptive statistics were produced for all tested individuals who have a result and then separately, comparing those who tested positive and negative for SARS-CoV-2. In order to take into account local epidemic dynamics, standard errors for the descriptive statistics were produced taking into account clustering at the municipality (county) level.

We estimated a mixed effects multivariate logistic regression to estimate the probabilities of being hospitalized, intubated (for those hospitalized) and of dying as a function of sex, age, comorbidities, health provider, whether the individual speaks an indigenous language, and quintiles of the proportion of the population living in poverty in the municipality (county) of residence. The model included municipality random effects.

Both descriptive analyses and regressions were conducted using Stata 15.0 (Stata Corp). As the model includes multiple comparisons, we estimated the false discovery rate using the Benjamini–Hochberg procedure by comparing each p-value to its Benjamini-Hochberg critical value, $(i/m)Q$, using 5% as Q [24].

For the percentage of the population living below the poverty line, we used the official multidimensional measure in Mexico that includes income and six measures of social deprivation (education, health, social security, housing, housing utilities, and food security) produced by Mexico´s National Council on Evaluation of Social Policy [25] which classifies municipalities by the proportion of individuals living in poverty.

## Results

Characteristics of the 1,378,002 individuals who were tested for SARS-CoV-2 in Mexico through September 16, 2020 are reported in Table 1, comparing those with positive and negative results (n = 654,858 and 723,144, respectively). From the pool of individuals tested, those

**Table 1. Sociodemographic and health conditions (% and 95% confidence interval) of Mexicans tested for SARS-CoV-2, total and by test result.**

| Sociodemographic characteristics | All tested | SARS-CoV-2 (+) | SARS-CoV-2 (-) | p value |
|---|---|---|---|---|
| Sex (Male) | 49.02 | 52.21 | 46.13 | <0.001 |
| | (48.60–49.44) | (51.80–52.63) | (45.59–46.67) | |
| Age (Average) | 44.11 | 46.07 | 42.33 | <0.001 |
| | (43.90–44.32) | (45.84–46.30) | (42.13–42.52) | |
| Indigenous | 0.88 | 1.01 | 0.76 | <0.001 |
| | (0.73–1.04) | (0.84–1.19) | (0.62–0.90) | |
| Ministry of Health | 65.13 | 58.36 | 71.25 | <0.001 |
| | (62.63–67.62) | (55.89–60.83) | (68.64–73.87) | |
| Social security | 31.91 | 38.50 | 25.94 | <0.001 |
| | (29.50–34.32) | (36.06–40.94) | (23.47–28.41) | |
| Private | 2.96 | 3.14 | 2.81 | 0.297 |
| | (1.90–4.03) | (2.27–4.01) | (1.52–4.09) | |
| Municipality poverty Q1 | 13.14 | 12.39 | 13.83 | 0.115 |
| | (6.78–19.51) | (6.36–18.42) | (7.06–20.60) | |
| Q2 | 54.46 | 54.04 | 54.84 | 0.573 |
| | (45.74–63.18) | (45.72–62.35) | (45.57–64.11) | |
| Q3 | 25.04 | 26.05 | 24.12 | 0.074 |
| | (17.92–32.16) | (19.10–33.00) | (16.71–31.53) | |
| Q4 | 6.61 | 6.66 | 6.56 | 0.756 |
| | (5.00–8.21) | (5.12–8.20) | (4.84–8.27) | |
| Q5 | 0.75 | 0.87 | 0.65 | <0.001 |
| | (0.55–0.96) | (0.63–1.10) | (0.47–0.84) | |
| Health conditions | All tested | SARS-CoV-2 (+) | SARS-CoV-2 (-) | p value |
| Obesity | 16.39 | 18.77 | 14.24 | <0.001 |
| | (15.84–16.95) | (18.23–19.32) | (13.68–14.80) | |
| Diabetes | 12.99 | 16.19 | 10.10 | <0.001 |
| | (12.57–13.42) | (15.76–16.62) | (9.73–10.46) | |
| Hypertension | 17.13 | 20.26 | 14.30 | <0.001 |
| | (16.53–17.74) | (19.62–20.91) | (13.80–14.80) | |
| Asthma | 2.86 | 2.58 | 3.11 | <0.001 |
| | (2.69–3.03) | (2.42–2.75) | (2.91–3.30) | |
| COPD | 1.41 | 1.53 | 1.29 | 0.009 |
| | (1.32–1.49) | (1.44–1.62) | (1.20–1.38) | |
| Chronic renal disease | 1.84 | 1.96 | 1.72 | <0.001 |
| | (1.73–1.94) | (1.86–2.06) | (1.59–1.85) | |
| Cardiovascular disease | 1.99 | 2.04 | 1.95 | 0.022 |
| | (1.90–2.08) | (1.95–2.14) | (1.84–2.05) | |
| Smoking | 7.47 | 9.68 | 8.63 | <0.001 |
| | (6.97–7.97) | (8.84–10.52) | (7.95–9.31) | |
| COVID-19 Outcomes | All tested | SARS-CoV-2 (+) | SARS-CoV-2 (-) | p value |
| Hospitalized | 17.38 | 25.06 | 10.42 | <0.001 |
| | (16.18–18.58) | (23.65–26.47) | (9.53–11.31) | |
| Intubated (of those hospitalized) | 15.53 | 18.14 | 9.84 | <0.001 |
| | (14.67–16.39) | (17.16–19.12) | (9.19–10.49) | |
| Death | 6.45 | 10.95 | 2.37 | <0.001 |
| | (5.96–6.93) | (10.32–11.58) | (2.13–2.61) | |
| Obs | 1,378,002 | 654,858 | 723,144 | |

who tested positive for SARS-CoV-2 were more likely to be male (52.21% vs. 46.13%, p < 0.001) and older (46.017 years vs. 42.33 years, p < 0.001). While the percentage of individuals who speak an indigenous language was low (0.88%), a higher proportion were SARS-COV-2 positive (1.01% vs 0.76%, p < 0.01). The proportion of individuals living in poverty at the municipality (county) level was similar between those who tested positive and negative.

In terms of health provider, 65.13% of individuals were treated at Ministry of Health (MoH) facilities, 31.91% at social security facilities and 2.96% by private providers. Among those who tested positive, 58.36% received care from the MoH, 38.50% from social security and 3.14% from private providers.

Individuals who tested positive to SARS-CoV-2 were more likely to have a diagnosis of obesity (18.77% vs 14.24%, p < 0.001), diabetes (16.19% vs 10.10%, p < 0.001), hypertension (20.26% vs 14.30%, p < 0.001), COPD (1.53% vs 1.29%) and diagnosis of chronic renal disease (1.96% vs 1.72%, p < 0.001) compared to those who tested negative, and were more likely to report smoking (9.68% vs 8.63%, p < 0.001). In contrast, they were less likely to report asthma (2.58% vs 3.11%, p < 0.001). There was no significant difference by diagnosis of cardiovascular disease (2.04% vs 1.95%, p = 0.071).

## Hospitalization

Being male was associated with an increased probability of being hospitalized (OR 1.66). Probability also increased with age, with an OR of 1.53 for those 30 to 39 years compared to 20 to 29 years, 2.99 for 40 to 49 years, 5.64 for 50 to 59 years, 10.95 for 60 to 69 years, 17.40 for 70 to 79 years, and 22.55 for those 80 years and older (Table 2).

The odds of hospitalization were 2.82 for those obese, diabetic, and hypertensive compared to those with none of those conditions, a higher probability that for any of those conditions alone. Those with COPD were more likely to being hospitalized (OR 1.42), while smoking alone was negatively associated with hospitalization (OR 0.95). Chronic renal disease was associated with a higher probability of hospitalization (OR 2.73) as well as cardiovascular disease (OR 1.15).

Those cared for by social security and private providers were more likely to be hospitalized (OR 3.13 & 2.81, respectively) compared to those from MoH facilities. Individuals who speak an indigenous language were more likely to be hospitalized (OR 1.64) compared to those who don´t. In terms of share of poverty at municipality level, the odds of hospitalization increase with the share of poverty which goes from OR 1.36 for individuals from municipalities with a share of poverty between 20% to 39% up to OR 3.25 for those from municipalities with a share of poverty of 80% and higher.

## Intubation

Being male was associated with an increased probability of being intubated (OR 1.26) as was age, with the probability increasing monotonically to age 70 to 79 (OR 2.77) (Table 2). It is lower for those 80 and above, but not significantly, despite an increase in mortality. That pattern is consistent with a reluctance to intubate patients over 80 at comparable levels of severity.

Being obese, diabetic and hypertensive increased the odds of intubation both as individual conditions and as comorbidities in a synergistic way: the odds ratio for intubation was 1.34 for those who were obese, diabetic & hypertensive compared to individuals with none of those conditions, higher than for any of these conditions alone.

Neither asthma, COPD nor smoking were related to the odds of intubation among the analyzed individuals. Chronic renal disease was positively associated with intubation (OR 1.15).

**Table 2. Odds (95% confidence interval) of hospitalization, intubation and death for individuals with COVID-19 based on comorbidities, demographics and socio-economic indicators[†].**

| | % | (1) Hospitalization | (2) Intubation | (3) Death |
|---|---|---|---|---|
| Sex (Male = 1) | 49.02 | 1.66*** | 1.26*** | 1.77*** |
| | (48.60–49.44) | (1.63–1.68) | (1.23–1.30) | (1.74–1.81) |
| Age 20 to 29 | 15.83 | 1.00 | 1.00 | 1.00 |
| | (15.42–16.26) | | | |
| Age 30 to 39 | 23.13 | 1.53*** | 1.30*** | 2.11*** |
| | 22.85–23.42) | (1.48–1.58) | (1.17–1.44) | (1.95–2.28) |
| Age 40 to 49 | 22.37 | 2.99*** | 1.73*** | 5.52*** |
| | (22.14–22.60) | (2.90–3.08) | (1.57–1.91) | (5.13–5.94) |
| Age 50 to 59 | 18.38 | 5.64*** | 2.21*** | 12.27*** |
| | (18.12–18.65) | (5.47–5.82) | (2.00–2.43) | (11.42–13.18) |
| Age 60 to 69 | 11.53 | 10.95*** | 2.71*** | 26.06*** |
| | (11.29–11.78) | (10.59–11.32) | (2.46–2.98) | (24.25–28.00) |
| Age 70 to 79 | 6.06 | 17.40*** | 2.77*** | 43.48*** |
| | (5.86–6.27) | (16.76–18.07) | (2.51–3.06) | (40.40–46.79) |
| Age 80+ | 2.69 | 22.55*** | 2.20*** | 60.53*** |
| | (2.57–2.81) | (21.51–23.64) | (1.98–2.44) | (56.02–65.42) |
| Indigenous Speaker = 1 | 1.01 | 1.64*** | 0.89* | 1.43*** |
| | (0.84–1.19) | (1.52–1.76) | (0.79–1.01) | (1.32–1.55) |
| Neither obese, diabetic nor hypertensive | 61.33 | 1.00 | 1.00 | 1.00 |
| | (60.48–62.18) | (1.00–1.00) | (1.00–1.00) | (1.00–1.00) |
| Neither obese nor diabetic. Hypertensive | 8.10 | 1.42*** | 1.16*** | 1.39*** |
| | (7.85–8.35) | (1.39–1.46) | (1.11–1.21) | (1.35–1.44) |
| Neither obese nor hypertensive. Diabetic | 5.71 | 2.28*** | 1.10*** | 1.82*** |
| | (5.56–5.87) | (2.22–2.34) | (1.05–1.15) | (1.76–1.88) |
| Not obese. Diabetic & hypertensive | 5.98 | 2.36*** | 1.09*** | 1.93*** |
| | (5.74–6.21) | (2.30–2.43) | (1.05–1.14) | (1.87–1.99) |
| Obese; neither diabetic nor hypertensive | 10.91 | 1.62*** | 1.30*** | 1.68*** |
| | (10.57–11.25) | (1.59–1.66) | (1.23–1.36) | (1.62–1.73) |
| Obese & hypertensive. Not diabetic | 3.35 | 1.86*** | 1.40*** | 1.95*** |
| | (3.21–3.48) | (1.80–1.93) | (1.31–1.48) | (1.87–2.03) |
| Obese & diabetic. Not hypertensive | 1.68 | 2.85*** | 1.32*** | 2.44*** |
| | (1.62–1.74) | (2.72–2.98) | (1.22–1.43) | (2.30–2.58) |
| Obese, diabetic & hypertensive | 2.76 | 2.82*** | 1.34*** | 2.48*** |
| | (2.63–2.88) | (2.71–2.92) | (1.27–1.42) | (2.38–2.59) |
| Neither asthma, COPD, nor smoker | 89.90 | 1.00 | 1.00 | 1.00 |
| | (88.54–89.47) | (1.00–1.00) | (1.00–1.00) | (1.00–1.00) |
| Smoking. Neither asthma nor COPD | 6.97 | 0.95*** | 1.03 | 0.97 |
| | (6.49–7.44) | (0.92–0.98) | (0.98–1.08) | (0.94–1.01) |
| COPD. Neither asthma nor smoker | 1.14 | 1.42*** | 1.07* | 1.35*** |
| | (1.08–1.21) | (1.34–1.50) | (0.99–1.16) | (1.27–1.42) |
| Smoking & COPD. Not asthma | 0.27 | 1.43*** | 0.96 | 1.12** |
| | (0.24–0.30) | (1.27–1.61) | (0.83–1.11) | (1.01–1.26) |
| Asthma. Neither COPD nor smoker | 2.28 | 0.94*** | 1.01 | 0.90*** |
| | (2.12–2.44) | (0.89–0.98) | (0.92–1.12) | (0.84–0.96) |

(*Continued*)

**Table 2.** (Continued)

| | % | (1) Hospitalization | (2) Intubation | (3) Death |
|---|---|---|---|---|
| Asthma & smoking. Not COPD | 0.18 | 1.03 | 0.86 | 0.96 |
| | (0.16–0.19) | (0.86–1.22) | (0.60–1.23) | (0.75–1.24) |
| Asthma & COPD. Not smoking | 0.08 | 0.97 | 0.98 | 1.07 |
| | (0.07–0.09) | (0.79–1.20) | (0.71–1.35) | (0.86–1.34) |
| Asthma, COPD & smoking | 0.03 | 0.48*** | 0.70 | 0.44*** |
| | (0.03–0.04) | (0.34–0.68) | (0.38–1.29) | (0.29–0.68) |
| Chronic renal disease | 1.96 | 2.73*** | 1.04 | 2.40*** |
| | (1.86–2.06) | (2.60–2.86) | (0.98–1.10) | (2.30–2.51) |
| Cardiovascular disease | 2.04 | 1.15*** | 0.98 | 1.06*** |
| | (1.95–2.14) | (1.10–1.20) | (0.92–1.05) | (1.02–1.12) |
| Ministry of Health | 58.36 | 1.00 | 1.00 | 1.00 |
| | (55.89–60.83) | | | |
| Social security | 38.50 | 3.13*** | 1.29*** | 2.91*** |
| | (36.06–40.94) | (3.08–3.18) | (1.25–1.33) | (2.86–2.97) |
| Private health provider | 3.14 | 2.81*** | 1.06 | 0.85*** |
| | (2.27–4.01) | (2.70–2.93) | (0.97–1.16) | (0.79–0.92) |
| **Quintiles of share of poverty** | | | | |
| Quintile 1 (0% to 19% of individuals are poor) | 12.39 | 1.0 | 1.0 | 1.0 |
| | (6.36–18.42) | | | |
| Quintile 2 (20% to 39% of individuals are poor) | 54.04 | 1.36** | 0.97 | 1.15 |
| | (45.72–62.35) | (1.01–1.84) | (0.79–1.20) | (0.94–1.41) |
| Quintile 3 (40% to 59% of individuals are poor) | 26.05 | 1.97*** | 0.84 | 1.39*** |
| | (19.10–33.00) | (1.48–2.62) | (0.69–1.04) | (1.14–1.69) |
| Quintile 4 (60% to 79% of individuals are poor) | 6.66 | 2.55*** | 0.79** | 1.60*** |
| | (5.12–8.20) | (1.92–3.39) | (0.64–0.97) | (1.32–1.95) |
| Quintile 5 (80% & + of individuals are poor) | 0.87 | 3.25*** | 0.98 | 1.95*** |
| | (0.63–1.10) | (2.40–4.41) | (0.76–1.25) | (1.56–2.43) |
| var(_cons[Municipality]) | | 2.39*** | 1.29*** | 1.33*** |
| | | (2.21–2.59) | (1.24–1.35) | (1.28–1.37) |
| Observations | | 631,821 | 158,938 | 631,821 |
| Number of groups | | 2,173 | 1,963 | 2,173 |

[†]Multiple comparisons controlled using a false discovery rate approach, following the Benjamini–Hochberg procedure comparing each p-value with its Benjamini-Hochberg critical value, (i/m)Q, using 5% as Q. All variables with a p value < 0.05 have also values < the Benjamini-Hochberg critical value.

*** $p < 0.01$,

** $p < 0.05$,

* $p < 0.1$

Being a patient in one of the social security institutions was associated with a higher probability of intubation (OR 1.29) compared to Ministry of Health (MoH) facilities. The probability of intubation was not different between the MoH and private providers. Compared to those from municipalities with the lowest share of poverty, individuals from municipalities with higher shares of poverty were less likely to be intubated; there was no association between intubation and speaking an indigenous language.

## Mortality

The probability of dying from COVID-19 was higher for males (OR 1.77) compared to females and increased with age: odds of death compared to individuals 20 to 29 were 2.11 for those 30 to 39, 5.52 for those 40 to 49, 12.27 for those 50 to 59, 26.06 for those 60 to 69, 43.48 for those 70 to 79 and 60.53 for those 80 and older. Speaking an indigenous language was associated with an increased probability of dying (OR 1.43) (Table 2).

The odds ratio for dying from COVID-19 was 2.48 for individuals with a diagnosis of obesity, diabetes and hypertension compared to those without any of those conditions. Being only obese (neither diabetic nor hypertensive) had an odds ratio of 1.68, for only diabetic it was 1.82, and for only hypertensive, 1.39.

Individuals with COPD were more likely to die from COVID-19 compared to those that were neither smokers, nor COPD, nor asthmatics with an OR of 1.35. Those with asthma but neither COPD nor smoking were less likely to die from COVID-19 with an OR of 0.90. Those with asthma, COPD and smoking had also a lower probability of dying of COVID-19 (OR 0.44). Individuals with chronic renal disease were more likely to die, with an odds ratio of 2.40.

Compared to individuals from Ministry of Health services, those from social security were more likely to die, with an OR of 2.91, while those from private providers were less likely to die, with an OR of 0.85.

Regarding share of poverty at the municipality level, compared to those with the lowest share of poverty, individuals from municipalities with a share of poverty between 40% to 59% had an increased probability of dying with an OR of 1.39, as well as those from municipalities with share of poverty between 60% to 79% and those from municipalities with a share of poverty of 80% and larger, with ORs of 1.60 and 1.95, respectively.

## Discussion

Given the high burden of NCDs in Mexico, in particular obesity, diabetes and hypertension, the results presented here represent a huge challenge for the county in terms of the current COVID-19 pandemic. Mexico is behind only the USA among OECD member countries in terms of prevalence of obesity among adults and has the highest rate in the OECD for diabetes [12, 26]. While the reported prevalence of these conditions among individuals tested for SARS-CoV-2 in Mexico is lower than the survey-based observed prevalence among the general population, it is similar to the self-reported prevalence of prior diagnosis of these conditions. In 2018 these were: 22.8% for obesity, 10.3% for diabetes, and 18.4% for hypertension [27].

As has been previously suggested, there is an urgent need to provide more complete data on the comorbidities associated with COVID-19 severity [5], in particular those NCDs that the evidence already suggests increase the probability of severe disease and death. For example, there are likely to be important differences by the severity of the comorbid conditions and/or by the type of treatment the patient is receiving.

In addition to the weighty burden of NCDs, Mexico faces another enormous challenge: social inequity. Wealth distribution in Mexico is one of the worst worldwide; Mexico is tied with Chile for the most unequal income distribution in the OECD [28]. Wealth inequity is the major driver of health inequity [29]. Poverty may be linked to higher risk of dying for COVID-19 not only due to the fact that poverty increases the probability of becoming infected because social distancing is more difficult or because the poor are at higher risk of developing NCDs, but also because individuals living in poverty have less access to public services, including health services, and those health services that provide care to them are on average of lower quality. Low-quality healthcare is an important cause of avoidable deaths in Mexico [30].

Mexicans who speak an indigenous language are five times as likely as those who do not to be living in extreme poverty [31]. Both speaking an indigenous language and living in a municipality (similar to a county in the USA) with a higher percentage of the population living in poverty were associated with increased mortality, in a stepwise fashion. There are many possible explanations (preexisting health status, delays in accessing care, and differential quality of care among the most important). Because the poor are also less likely to access the social security or private health systems, this compounds their mortality risk [31]. Our results are consistent with similar studies reported for other countries: in Brazil, COVID-19 mortality has been reported to be associated with minority ethnic background and region of residence [19]. Similarly, in the UK there is evidence of the pandemic disproportionally affecting minority ethnic communities [32].

Mexico requires, as does the rest of the world, a *precision public health* approach, one that take advantage of available data to design a response that takes into account population characteristics as determinants of risk.

Lower mortality for those individuals with asthma, those who smoke, and those with asthma, COPD and who smoke may be related to the fact that individuals from these groups may be more likely to seek care earlier and to be tested sooner because they are perceived to be at-risk. However, the results are also consistent with other studies that have reported lower risk among smokers [33, 34]. Further analyses and more complete data would be required to better describe these relationships.

This study has an important limitation. It is a study of COVID-19 patients in Mexico and we are not able to correct for disease severity. Thus, when differences are observed among institutions or groups, the difference could be due to differential thresholds for seeking any health care, for hospitalization or for intubation. For example, the higher mortality in the Ministry of Health hospitals compared to the private sector could imply that the same patient, depending on where they were hospitalized, would have a higher chance of dying in an MoH hospital. However, it could equally well represent that a patient needs to be more severely ill to be hospitalized in a MoH hospital than in the private sector. Thus, one must be careful not to assume causality. That said, the significantly higher morbidity and mortality associated with co-morbid conditions probably underestimates their true effect, to the extent that physicians are likely to have a lower severity threshold for recommending hospitalizing for a patient with comorbidities. Additionally, as the national protocol mandates testing for all individuals with severe respiratory conditions who present to health facilities as well as a proportion of those with mild respiratory conditions, results may underestimate how poverty is associated with COVID-19 prognosis as those living in poverty have less access to health services. In terms of association with NCDs, this testing protocol may also underestimate associations, as those with mild respiratory conditions will have a better prognosis. Our socioeconomic indicator, share of poverty at the municipality level, reflects not the condition of the individual but that of the county where she or he lives.

Mexico's response to COVID-19 is stymied by a synergistic double challenge: raging NCDs and extreme social inequity. The response to the current pandemic must take both of them into account both to be effective and to ensure that the burden of COVID-19 not fall even more disproportionately on those who are already disadvantaged.

## Author Contributions

**Conceptualization:** Juan Pablo Gutierrez, Stefano M. Bertozzi.

**Data curation:** Juan Pablo Gutierrez.

**Formal analysis:** Juan Pablo Gutierrez.

**Methodology:** Juan Pablo Gutierrez.

**Writing – original draft:** Juan Pablo Gutierrez.

**Writing – review & editing:** Stefano M. Bertozzi.

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
