## [Decision Letter · Decision Letter 0]

29 Jun 2020

PONE-D-20-15813

Non-communicable diseases and inequalities increase risk of death among COVID-19 patients in Mexico

PLOS ONE

Dear Dr. Gutierrez,

Thank you for submitting your manuscript to PLOS ONE. After careful consideration, we feel that it has merit but does not fully meet PLOS ONE’s publication criteria as it currently stands. Therefore, we invite you to submit a revised version of the manuscript that addresses the points raised during the review process.

A **rebuttal letter** that responds to **EACH** point raised by the academic editor and reviewer(s). You should upload this letter as a separate file labeled 'Response to Reviewers'.A **marked-up copy** of your manuscript that highlights changes made to the original version. You should upload this as a separate file labeled 'Revised Manuscript with Track Changes'.An **unmarked version** of your revised paper without tracked changes. You should upload this as a separate file labeled 'Manuscript'.

We look forward to receiving your revised manuscript.

Kind regards,

Brecht Devleesschauwer

Academic Editor

PLOS ONE

Additional Editor Comments:

In your revision note, please include EACH of the reviewer comments, provide your reply, and when relevant, include the modified/new text (or motivate why you decided not to modify the text). Note that failure to do so may result in a rejection of the manuscript.

Journal Requirements:

2. In your Methods section, please provide additional information about the methodology of your study. Please ensure you have provided sufficient details to replicate the analyses such as: a) the date on which the dataset was accessed, b) a description of all variables included in the analysis, and explanation on how these were categorised. Moreover, please refer to the specific statistical analyses performed as well as any post-hoc corrections to correct for multiple comparisons. If these were not performed please justify the reasons. Please refer to our statistical reporting guidelines for assistance (https://journals.plos.org/plosone/s/submission-guidelines.#loc-statistical-reporting). Additionally, please ensure you have thoroughly discussed any potential limitations of this study within the Discussion.

Reviewers' comments:

Reviewer's Responses to Questions

**Comments to the Author**

1. Is the manuscript technically sound, and do the data support the conclusions?

Reviewer #1: Yes

Reviewer #2: Partly

Reviewer #3: Yes

Reviewer #4: Yes

2. Has the statistical analysis been performed appropriately and rigorously? 

Reviewer #1: No

Reviewer #2: No

Reviewer #3: Yes

Reviewer #4: Yes

3. Have the authors made all data underlying the findings in their manuscript fully available?

Reviewer #1: Yes

Reviewer #2: No

Reviewer #3: Yes

Reviewer #4: Yes

4. Is the manuscript presented in an intelligible fashion and written in standard English?

Reviewer #1: Yes

Reviewer #2: Yes

Reviewer #3: Yes

Reviewer #4: Yes

5. Review Comments to the Author

Reviewer #1: This is an important study sharing data on the COVID-19 experience in Mexico. The findings highlight the burden of non-communicable disease in the country, and the subsequent implications for the COVID-19 pandemic.

My comments on the manuscript are as follows:

I used the link in the document to source the original data and downloaded the csv file. The column headings are in Spanish, and took a little time to figure out, and I did not immediately understand the key. However, initial look at the data reassured me that the results presented in the text reflect the data.

In general, the manuscript is brief, and so I did have some questions on the methods. It would be important to have a clearer understanding of the data source. Who is included, who might be excluded? Are there any data verification steps? As the data are from an administrative source, it is important the authors share as much information as possible about the reliability of the information. I am not familiar with Mexican administrative datasets.

I am struggling to understand the 'pool' of tested patients adequately. What were the country criteria for testing? Where were tests conducted? Were these limited to hospital settings? This may of course be very relevant to the subsequent indigenous language analyses.

The logistic regression models were simple - exploring risks of hospital admission, intubation or death. The models use outcomes in a binary manner - but my evaluation of the dataset led me to believe dates were included for death, so survival models would have been an option. Was this considered?

In the logistic regression, I would be keen to understand the detail of the models controlling for hospitalisation? Surely it would have been more appropriate to analyse intubation only amongst hospitalised patients? Maybe this was the case - but I was not sure.

Could the authors justify the choice of confounders (were these simply reflective of available data)?

The results tables need clearer labelling. I presumed age was mean +/- SD... but this should be explicitly stated. Similarly, variables like 'obesity' need defining clearly.

The rate of admission in the non-sars-cov2 group struck me as high. Perhaps the authors could add the sensitivity of their testing kits in Mexico?

The second table should ideally present the odds ratios alongside 95% confidence intervals. There is an extraordinary amount of hypothesis testing being done, and I would be more comfortable with fewer p values presented!

Along the same lines, there appear to be some unusual interactions, possibly with effect modification, between asthma, COPD and smoking. My instinct these are confounded by age. I would like to know how the authors explored these associations.

Ideally I would have liked to see the logistic regression data accompanied by absolute numbers, to be able to appreciate absolute v relative risks.

The discussion is brief (I appreciate this is a concise report). However, my sense is that the really novel aspects of the results are the descriptives of the cohort in Mexico in terms of burden of NCD, plus the work around deprivation and indigenous language. The observations around age and comorbidity and their relationship with SARS-CoV-2 outcome are well described now.

There are some minor edits required for typographic errors, but overall the language and style are good.

Reviewer #2: Dear Author,

Thank you for the manuscript, as it possibly adds something to the existing literature, presenting risk factors for hospitalization, intubation and death in the Mexican context. A topic of interest that could be better explored in this manuscript is the effect of socioeconomic inequalities in these outcomes.

I think that this letter could be greatly improved with better background and methods, mainly by considering previously existing articles and by clarifying and improving the methodological approach.

It is noteworthy to mention that a similar (smaller) study was also published, without peer review process, (doi: 10.1101/2020.05.11.20098145), although it did not consider any socioeconomic indicator regarding poverty.

Abstract

The abstract should be re-written. There are a lot of different messages and outcomes; The rationale and conclusions should be more aligned with the main outcomes.

Introduction

According to the main outcomes presented, it seems that the study aims to assess risk factors, including a socioeconomic indicator, for (1) hospitalization, (2) intubation and (3) death. Having said this, the goals should be redefined accordingly.

Several already published studies with similar aims should also be referred in this section. Some examples (doi): 10.1136/bmj.m1985, 10.1080/13685538.2020.1774748, 10.1101/2020.05.06.20092999, 10.1002/jmv.26050 or 10.1016/j.jinf.2020.04.021. Other literature on inequalities and COVID-19 could also be considered.

Methods and Results

The sample considered in this study should be “patients with COVID-19 (confirmed cases with SARS-CoV-2)”.

Why only consider adults as 20 years old or older?

There is no need to compare patients that tested positive and with those that tested negative for SARS-CoV-2 as it does not add information. Thus, the first part of the Results section can be deleted. Table 1 (descriptive analysis) should be reviewed considering the correct outcomes.

Reference to “co-morbidities” or “comorbidities” should be uniformed throughout the manuscript.

Information on data sources and quality should be clearly stated, as well as how missing data was handled (even if in supplementary material).

Regarding inequalities analysis, other methods could be considered. For example, why consider 50% on poverty as the cut-off instead of quintiles?

To estimate causation, other options might be considered or discussed in the Discussion (as limitations); e.g. directed acyclic graphs. Were all the analyzed variables included in the regression analyses? Any kind of univariate analysis firstly? For the logistic regression, 95% CI must be presented.

Table 2 – It should be self-explanatory. 95% CI should be presented. The presentation could also be improved. Outcomes should be (1) hospitalization, (2) intubation and (3) death.

Discussion

Discussion should be more focused on the aforementioned aims (risk factors, including socioeconomic, for these outcomes) of this study, including comparison with the existing literature.

Other limitations should be clearly presented (e.g. ecological fallacy of considering municipalities as socioeconomic indicators instead of individual information).

Reviewer #3: In this paper, using data from the national epidemiological surveillance system, authors analyzed individuals with a positive result to estimated odds of hospitalization, intubation and death, based on pre-existing non-communicable diseases and socioeconomic indicators.

The analysis is pertinent and appropriate to draw the attention of the authorities of a country, whose health system will face important challenges in the severity of comorbid conditions and in the type of treatment that patients should receive.

Minor comments:

Reference 6 is neither complete nor in the journal format. Maybe it refers to the reference:

Barquera S, Campos-Nonato I, Hernández-Barrera L, Pedroza A, Rivera-Dommarco JA. Prevalencia de obesidad en adultos mexicanos, 2000-2012 [Prevalence of obesity in Mexican adults 2000-2012]. Salud Publica Mex. 2013;55 Suppl 2:S151-60. Spanish. PMID: 24626691.

Some typographical errors are identified in reference 12. Correct please.

Reviewer #4: Gutierrez and Bertzozzi provide a descriptive analysis of the COVID-19 pandemic in terms of the comorbidities and particularities of care and poverty amongst Mexicans suspected for COVID-19. The paper is well-written and the results are confirmatory of previously published studies (doi:10.1016/j.orcp.2020.06.001 , doi:10.1210/clinem/dgaa346/5849337, doi:10.1101/2020.05.11.20098145). A novel finding of this work is the association of poverty with COVID-19 outcomes; this confirms findings of a previous study which linked the social-lag index to increased risk of adverse outcomes for older adults in Mexico (doi:10.1093/gerona/glaa163). The authors are encouraged to contextualize their findings in light of these previous works and address the following concerns:

1. Authors incorporate multilevel data regarding poverty status and comment that "In order to take into account local epidemic dynamics, estimations assumed that observations are clustered at the municipality level"; however, no specific description of the statistical technique utilized is being referred. Was this a mixed effects logistic model? Please specify.

2. In the discussion section authors refer that severity of disease cannot be specified given available data but that is inaccurate. Previous definitions of severe COVID-19 have been based on composite outcomes of death, ICU admission and intubation, all of which are available for the Mexican Health Ministry Dataset (doi:10.1001/jamainternmed.2020.2033). Please comment on this and consider including an outcome related to severe COVID-19.

3. A main concern of the present work is the definition of inequities. Whilst inarguably, poverty is a main driver of social inequity, there are additional factors which condition in equity and are better captured by other metrics measured by CONEVAL (http://sticerd.lse.ac.uk/dps/case/cp/CASEpaper205.pdf, https://www.coneval.org.mx/Medicion/IRS/Paginas/Que-es-el-indice-de-rezago-social.aspx). Authors are concouraged to either consider additional multidimensional metrics to be able to fully capture inequality or refer to their findings in relation to poverty.

4. Authors should comment on how the approach to case tracing and confirmation in Mexico might bias their results towards high-risk cases. Given that most cases which are being tested are symptomatic cases from key risk groups, interpretation of findings should be framed in this sense.

5. In the statistical analysis section it is asserted that "In addition to demographic characteristics and comorbidities, we also included state of residence to control for variations related to differences between states". This contradicts the previous comment that obervations were clustered to accunt for local dynamics. Simply adjusting for state of occurence controls some variability but the ideal model would incorportate this as a random effect within a mixed effects framework. These differences are relevant and a mixed effect model would effectively control these variabilities.

6. PLOS authors have the option to publish the peer review history of their article (what does this mean?). If published, this will include your full peer review and any attached files.

Reviewer #1: **Yes: **James Galloway

Reviewer #2: No

Reviewer #3: No

Reviewer #4: No

---

## [Author Response · Author response to Decision Letter 0]

28 Jul 2020

Please see attached file with responses to each comment.

---

## [Decision Letter · Decision Letter 1]

15 Sep 2020

PONE-D-20-15813R1

Non-communicable diseases and inequalities increase risk of death among COVID-19 patients in Mexico

PLOS ONE

Dear Dr. Gutierrez,

Thank you for submitting your manuscript to PLOS ONE. After careful consideration, we feel that it has merit but does not fully meet PLOS ONE’s publication criteria as it currently stands. Therefore, we invite you to submit a revised version of the manuscript that addresses the points raised during the review process.

A **rebuttal letter** that responds to **EACH** point raised by the academic editor and reviewer(s). You should upload this letter as a separate file labeled 'Response to Reviewers'.A **marked-up copy** of your manuscript that highlights changes made to the original version. You should upload this as a separate file labeled 'Revised Manuscript with Track Changes'.An **unmarked version** of your revised paper without tracked changes. You should upload this as a separate file labeled 'Manuscript'.

We look forward to receiving your revised manuscript.

Kind regards,

Brecht Devleesschauwer

Academic Editor

PLOS ONE

Additional Editor Comments (if provided):

Thank you for addressing the reviewer and editorial comments. Reviewer #2 raised some additional minor issues, which could be addressed in a final revision round.

Reviewers' comments:

Reviewer's Responses to Questions

**Comments to the Author**

1. If the authors have adequately addressed your comments raised in a previous round of review and you feel that this manuscript is now acceptable for publication, you may indicate that here to bypass the “Comments to the Author” section, enter your conflict of interest statement in the “Confidential to Editor” section, and submit your "Accept" recommendation.

Reviewer #2: (No Response)

Reviewer #4: All comments have been addressed

2. Is the manuscript technically sound, and do the data support the conclusions?

Reviewer #2: Yes

Reviewer #4: Yes

3. Has the statistical analysis been performed appropriately and rigorously? 

Reviewer #2: I Don't Know

Reviewer #4: Yes

4. Have the authors made all data underlying the findings in their manuscript fully available?

Reviewer #2: Yes

Reviewer #4: Yes

5. Is the manuscript presented in an intelligible fashion and written in standard English?

Reviewer #2: Yes

Reviewer #4: Yes

6. Review Comments to the Author

Reviewer #2: Dear Author,

Thank you for the reviewed manuscript and for the answer to the reviewers’ comments.

I would reinforce that it would be useful to mention that a similar (without peer review process) study was also published (doi: 10.1101/2020.05.11.20098145), although with a small number of patients and not consider any socioeconomic indicator regarding poverty.

Please do not use inequality and inequity interchangeably.

Please do a proof-reading once again.

Introduction

“Other positively associated conditions that are also major contributors to the global burden of disease [7]” – it seems that it is not referring COVID-19 severity (?)

“Diabetes is the single largest contributor to disease burden in Mexico, with a prevalence of about 15% of the adult population. Hypertension is also very important, with a prevalence of 30%.” – please use references to support this

“The triad of NCD, inequality and COVID doesn’t occur in the context of a high-functioning health care system, but rather one that is characterized by fragmentation, under-funding, lax regulatory oversight and a high proportion of poorly trained health personnel. The unfortunate result is low average health care quality which translates into unacceptably high mortality rates for severe COVID-19 disease.” – this triad exist in every countries with NCD, inequalities (depending on the cut-off) and with COVID-19, which correspond to almost every countries in the world. Please rephrase.

Please reformulate the aim of the study as suggested before.

Methods and Results

Missing data – It would be important to clearly state this with more detail.

Have the quintiles such an exact distribution (i.e. 20%, 40%, …)?

Table 2 – It seems still quite confusing, e.g. stating “probability” instead of “Odds”.

Reviewer #4: Thank you for addressing my comments, the paper is now ready for publication. This study adds to other analyses published using the Mexican registry database.

7. PLOS authors have the option to publish the peer review history of their article (what does this mean?). If published, this will include your full peer review and any attached files.

Reviewer #2: No

Reviewer #4: No

---

## [Editor Report · Decision Letter 2]

28 Sep 2020

Non-communicable diseases and inequalities increase risk of death among COVID-19 patients in Mexico

PONE-D-20-15813R2

Dear Dr. Gutierrez,

We’re pleased to inform you that your manuscript has been judged scientifically suitable for publication and will be formally accepted for publication once it meets all outstanding technical requirements.

Kind regards,

Brecht Devleesschauwer

Academic Editor

PLOS ONE
---

## [Editor Report · Acceptance letter]

30 Sep 2020

PONE-D-20-15813R2 

Non-communicable diseases and inequalities increase risk of death among COVID-19 patients in Mexico 

Dear Dr. Gutierrez:

I'm pleased to inform you that your manuscript has been deemed suitable for publication in PLOS ONE. Congratulations! Your manuscript is now with our production department. 

Kind regards, 

on behalf of

Prof. Dr. Brecht Devleesschauwer 

Academic Editor

PLOS ONE